# Children’s Understanding of Informed Assents in Research Studies

**DOI:** 10.3390/healthcare9070871

**Published:** 2021-07-10

**Authors:** Hortense Cotrim, Cristina Granja, Ana Sofia Carvalho, Carlos Cotrim, Rui Martins

**Affiliations:** 1Nursing Department, Universidade Atlântica, Fábrica da Pólvora de Barcarena, 2730-036 Barcarena, Portugal; 2Anaesthesiology Department, Centro Hospitalar Universitário São João, 4200-319 Porto, Portugal; cristinagranja@hotmail.com; 3Associated Teacher with Aggregation, Institute of Bioethics, University of Portugal, 1649-023 Lisboa, Portugal; acarvalho@porto.ucp.pt; 4Cardiology Department, Hospital da Cruz Vermelha, 1549-008 Lisboa, Portugal; carlosadcotrim@hotmail.com; 5Centro de Estatística e Aplicações da Universidade de Lisboa (CEAUL), Department of Statistics and Operational Research, Faculdade de Ciências, Universidade de Lisboa, 1549-008 Lisboa, Portugal; rmmartins@fc.ul.pt

**Keywords:** informed assent, children, comprehension capacity, decision-making ability

## Abstract

The assent procedure reflects an effort to enable the minor to understand, to the degree they are capable of, what their participation in the decision making process would involve. Aims: To evaluate the minors’ ability to understand the information provided to them when obtaining assent and to evaluate the opinion of the parents regarding the importance of asking the child’s assent. Methods: The sample included a total of 52 minors aged between 10 and 17 years who underwent exercise echocardiogram. The Quality of Informed Consent is divided into two parts: Part A was used to measure objective understanding and part B to measure subjective understanding. Results: The results show that the minors have a high capacity to understand the information given to them when asking for assent. A positive relationship was found between the two parts of the questionnaire. No statistically significant relationship was found between age and sex and part A and part B or between both age groups (<14 years old and ≥14 years old) and the measure. In the case of the parents, 96.6% of parents consider assent as an advantage for the child’s acceptance of health care. The opinion of the parents is not related to the age, sex or level of schooling. Conclusion: Minors showed a substantial level of understanding regarding the information provided to them. The parents considered the implementation of assent fundamental to the child’s acceptance of health care.

## 1. Introduction

Based on international ethical standards, the literature argues that minors have the right to be heard and to give their opinion, which should be taken into consideration progressively, according to their age, degree of maturity and discernment. Minors are thus increasingly considered to have rights and the capacity for self-determination as expressed in the United Nations Convention on the Rights of the Child of 20 November 1989 [1].

However, most parents and practitioners argue that the great majority of children will experience limitations in decision making. To account for these limitations, the term “informed assent” introduces a broader view on the capacity of decision making, recognizing that children to some extent should participate in decision making and that they can agree with treatment or participation in research without precisely understanding all of its consequences or without complying with all aspects of full consent [2].

This study appears as a method of deepening ethical knowledge regarding the recognition for the decision making capacity of minors concerning their participation in research studies, based on the notion of human vulnerability and the imperative need to create mechanisms that guarantee the protection of children to ensure respect for their dignity as human beings and care for their best interests.

It is also considered that this study can contribute to the enhancement of the health professionals’ approach to vulnerable populations, in particular children, in the design of strategies conducive to their empowerment and involvement in the decision making process.

The literature notes the importance of asking children for their assent, especially when they are enrolled in research. However, in order to comply with the best practices, assent should be requested whenever it is necessary to make a decision involving the health and well-being of the minor. In this study, we agreed with this assumption and argued that in research the assent is of utmost importance because frequently children do not benefit directly from the results obtained.

It is ethically and legally accepted that, in addition to the agreement of the parents regarding the inclusion of their children in medical programs, the minor should also express his willingness to participate, i.e., he must give his assent [3]. This request for assent represents a challenge related to the ability of minors to understand and make decisions since this is not a fixed phenomenon but a process that matures with time and experience. It should be noted that age alone does not indicate the ability of a child to understand, considering that factors such as knowledge, health status, anxiety, values, cultural, family, and religious contexts, for example, play an important role in their ability to understand and make decisions [4]. According to the authors, assent is about respecting children’s developing capacity, which means assisting them in understanding their condition and treatment at a developmentally appropriate level and involving them in appropriate decision making tasks [4]. 

As such, all aspects of the assent process should be clarified to avoid possible doubts about what is being questioned and to what extent the minor’s response will be respected [5]. These assumptions require the practitioner to master a set of high-quality communication skills due to the need to present an individualized communication appropriate to the child’s developmental stage, which constitutes personalized assent [6,7].

Technically, the assent is a document that explains to the child in the language she/he can understand the essence of what is planned in the research, as well as the fact that she/he can say ‘no’ or can change his/her mind midway through the research [3]. Accordingly, ‘Assent’ is a term used to express the willingness to participate in research by persons who are, by definition, too young to give informed consent but who are old enough to understand the proposed research in general, its expected risks and possible benefits and the activities expected of them as subjects. However, assent by itself is not sufficient. If assent is given, informed consent must still be obtained from the subject’s parents or guardian.

“Informed consent” is the voluntary agreement of an individual or his or her authorized representative who has the legal capacity to give consent and who exercises free power of choice without undue inducement or any other form of constraint or coercion to participate in research. The individual must have sufficient knowledge and understanding of the nature of the proposed research, the anticipated risks and potential benefits and the requirements of the research to be able to make an informed decision [8].

Therefore, child assent must be linked to a protective mechanism such as parental permission, even if the requirement for parental permission has been waived. The linked requirement for parental permission means that we do not need to burden child assent with the same informational and decision making standards as adult informed consent [9].

Regarding the importance of parents, the American Academy of Pediatrics Committee on Bioethics states that decision making by children and adolescents is usually influenced by their parents’ point of view and may not be entirely voluntary or autonomous. Unless there is significant coercion perceived by clinicians, this situation is not unacceptable because medical decision making cannot and should not occur in a vacuum isolated from all other concerns. Medical decision making is not a discrete event but evolves over time among the health care team, family and pediatric patient as new information becomes available [10]. 

Thus, based on ensuring compliance with the ethical standards in force, the assessment of the decision making capacity of the child should include the following: a review of the legal context consistent with the principles of the Convention on the rights of the child; the existence of an empathic relationship between the child and the team/researcher; respect for the child’s abilities and stage of development; the inclusion, if relevant, of relatives, experts, teachers and/or health and social professionals with their consent; the control of any type of coercion or other social forces that may influence their decision and, finally, the existence of a deliberative evaluation on the criteria of the adolescent’s decision making process [11,12].

The Teachers College Institutional Review Board of Columbia University states that the assent procedure should reflect a reasonable effort to enable the child (youth or adolescent) to understand, to the degree they are capable, what their participation in research involves. Assenting also involves researchers actively observing verbal and non-verbal cues that the youth does not want to participate (e.g., crying, fussing, throwing a tantrum, hesitation, distraction, discomfort, etc.). Researchers should pause (or stop) the study if they observe any signs of resistance the youth may express [13].

In addition to these aspects, the UK Royal College of Pediatrics stipulates “school-age” as a milestone from which the child is involved in decision making. Additionally, in the USA, the National Commission for the Protection of Human Subjects of Biomedical and Behavioural Research advocates the involvement of minors from the age of seven years old, which is a decision that is also supported by the American Academy of Pediatrics [14].

Some studies argue that the lack of involvement can have negative consequences, such as increased fears and anxieties, reduced self-esteem, depersonalization and lack of preparation for the procedures [15]. In the opinion of Coyne et al. [15], children who are not involved may assume that their opinions are not important or relevant and may not seek to share them in the future.

In this sense, Rippen [16] argues the importance of being aware that children have the right to the best possible medical care, the best information and education in comprehensive language suited to their age and a patient-oriented approach. If these aspects are considered, children generally feel better when they become involved in health care decisions and emotional problems and disorders in the development of the child can be prevented. However, Baines (2011) argues that assent is emphasized in research but is largely ignored in medical treatment for children. In this case, the child’s medical treatment may proceed with consent from the parents but the child’s participation in research requires both consent from the parents and the assent of the child and this may result in a practical problem if the parent and child provide incompatible responses when asked to consent and assent [17].

On the other hand, a double consent procedure will do justice to both developmental aspects of children and the specific characteristics of the parent-child dyad. The parental role offers extra protection by creating the context for the child’s competent decision making and by facilitating the child’s long-term autonomy. In general, the perspective and attitudes of the adults (both parents and clinician) towards the child may be an important predisposing factor in order to stimulate the highest competence in the child [18].

We must also consider the balance between the autonomy of the child versus the approach based on her best interest, the legal aspects versus the ethical and psychological aspects, the will of the child versus the will of the family and approaches that emphasize assent versus those that emphasize consent given by the child [19]. It should be noted that for a complete assent process, certain key elements must be considered, namely the explanation given by the participant about his understanding of the study, the risks and benefits involved, the alternatives to his participation, the potential consequences of participating or not, his understanding of the voluntary nature of his participation and the possibility of withdrawing from the study at any time [6].

However, even though the assent may empower the child to the extent that they are capable, Unguru [4] argues that parents should decide to what extent the child is in fact capable. In the point of view of Breeuwesma and van Geert [2], the easiest method to solve this problem is to accept parents or other caregivers of the child-patient as the main actors when it comes to making medical decisions. Regardless, the Belmont Report states that the requirement for parental permission and child assent is an application of utmost importance when regarding the general principle of respect for all persons [20].

Therefore, if the patient is a child, the family play an extra important role if the goal is to provide child and family centered care. In the opinion of Rippen [16], information sharing and communication with the patient and their loved ones on the decision making process are the key elements in applying true family-centered care. Furthermore, this author states that family integrated care not only means children and parents should be at the center but also actively involved in the care process, in decision making and regarded as equals.

The ruling of the parents is very important in this process because some children prefer a passive role in shared decision making because they are too ill or distressed by the treatments. Likewise, some children prefer to hear information from their parents especially if it is ‘bad’ news or, for example, about treatment side effects [21].

Children’s behavior varies constantly in terms of their involvement in the shared decision making process. Therefore, health professionals need to be attentive to the children’s need to ask questions by themselves. Children in the study of Beresford and Sloper [22] described how their parents dominated consultations, making it difficult for them to contribute to the interaction. Some children also described trying to ask questions to health professionals and being told to stay quiet by the parent. This points to the importance that health professionals need to take every opportunity to talk with children about their hopes and worries, their wants and needs, their ideas, goals and their decisions [2].

In summary, measuring children’s competence to make health care decisions raises a set of problems because decision making competence does not increase linearly with age but can vary greatly between moments and contexts. In this case, health professionals need to look at every child individually and accept that medical decision making is an ongoing process that varies depending on many factors. Therefore, considering the child’s best interest, parents should be involved in decision making and, alongside children, should act as collaborators with shared interests.

The purpose of this study was to evaluate the comprehension capacity of minors regarding the information given to them and to ultimately help health professionals better understand the importance of including children in the decision making and, at the same time, attempting the implementation of an informed assent model in Portugal’s pediatric health care. The aims of the study are as follows: (a) adapt the Quality of Informed Consent Questionnaire (QuIC) for a pediatric population; (b) evaluate the ability of children to understand the information made available to them regarding the request for assent to perform the exercise echocardiogram and for their inclusion in the study; (c) develop an informed assent model for pediatric population; (d) evaluate the opinion of their parents related to the child’s importance of assent to better accept the medical procedure.

## 2. Materials and Methods

### 2.1. Study Design

This is an exploratory, cross-sectional, descriptive and correlational study.

### 2.2. Sample Selection Inclusion

The convenience sample of this study is constituted 52 children who required an exercise echocardiogram (90% of the children that needed an exercise echocardiogram between August 2018 and August 2019 in Portugal). The reason for choosing these participants is their health condition as they are healthy children who present only some symptoms during the practice of physical exercise, which guarantees us the reduction in other confounding factors. The sample also includes the 52 parents to evaluate their opinion related to the importance of the implementation of the informed assent for their child.

The inclusion criteria is as follows: age between 10 years and 17 years; need to perform stress echocardiography as a diagnostic tool; be conscious and present cognitive ability to perceive the information provided, even if there was a need to adapt it to their intellectual level.

### 2.3. Consent Procedures

The study was approved by the Ethics Committees of three medical centers: Hospital Cruz Vermelha Portuguesa (ID: CIEE; 02, 2018, hl; Appendix A), UCARDIO (COMISSÃO DE ÉTICA PARA A SAÚDE DA SPAMEDIC, LDA., No. 1; 2018; Appendix A) and Hospital Particular do Algarve (ID No. 2/2018; Appendix A) where the data were collected. After these approvals, the parents and the children were carefully informed by the doctor who performed the exercise echocardiogram about the study and the doctor invited them to participate. The doctor also informed the children and their parents about the medical procedure by clarifying all the doubts. After that, informed consent was obtained from all subjects involved in the study. Thus, two written informed consents were requested from parents: one to allow children to perform the exercise echocardiogram and another to allow them to be involved in the study sample. Two written informed assents were also requested from children: one to perform the exercise echocardiogram and another to be included in the study sample.

The study complied with the ethical standards inherent to the Helsinki protocol. All data used were coded to be accessed only by the principal researcher and the duty of confidentiality and anonymity was met. Participation in the entire study was voluntary and informed.

### 2.4. Data Collection Procedures

The data were collected by the doctor before the performance of the exercise echocardiogram and after all the information considered as necessary to perform the medical procedure was given to the parents and children. Children were asked to answer the questionnaires independently and without help from their parents. We used an adaptation of the Quality of Informed Consent (QuIC) (Appendix A) constituted of two parts: Part A which evaluates the understanding of objective information; and part B which evaluates the understanding of subjective information developed by Joffe et al. [23].

This is a brief questionnaire originally developed to measure subjects’ actual (objective) and perceived (subjective) understanding of consent procedures by adult cancer patients participating in clinical trials. The questionnaire was translated to the Portuguese language by an official translator and the language was adapted to children.

Objective information refers to the knowledge of study procedures and their benefits and risks shared by the doctor prior to asking for assent; subjective information refers to the understanding of the possible impacts that participating in the study may have on children.

To evaluate the internal consistency of this measure, we calculated Cronbach’s alpha separately for part A and part B. For part A and B of QuIC, the values obtained were 0.68 and 0.77, respectively. The test-retest reliability of the original tools was considered good in the literature with intraclass correlation coefficients of 0.66 for tests of objective understanding and 0.77 for tests of subjective understanding [23].

Part A of the QuIC is composed of 17 statements and each has three response options: Disagree, with score 0; I am not sure, with a score of 1; and I agree, with a score of 2.

### 2.5. Data Analysis Procedures

Version 23 of SPSS (Statistical Package for the Social Sciences, Armonk, NY: IBM Corp.) software was used to perform the data analysis and the data were introduced according to a pre-established coding system to allow the identification of each variable.

Descriptive statistics with absolute and relative frequencies and measures of central tendency and dispersion were used to characterize the sample. Welch’s *t*-test was used to study the differences between the mean values of a variable relative to two independent groups. Pearson’s correlation coefficients were determined and analyzed to study the existence of linear correlations between quantitative variables.

## 3. Results

### 3.1. Sociodemographic Characteristics of Minors

We found that the age of children ranged between 10 and 17 years, with a mean age of 13.83 years and a standard deviation of 2.15 years.

We divided the ages into two classes: <14 years old and ≥14 years old. We found that 38.46% of the sample was in the age group <14 years old and 61.54% in the age group ≥14 years old. Regarding sex, 71.15% of the minors were male and 28.85% were female.

Crosstabulations between children’s age (<14 and ≥14) and gender (male/female) revealed that 11.5% (*n* = 6) of children were female and 26.9% (*n* = 14) were male under 14 years of age; 17.3% (*n* = 9) were female and 44.2% (*n* = 23) were male equal to or over 14 years of age. 

### 3.2. Sociodemographic Characteristics of Parents

As for the age of parents, we found that it varied between the age group of 30–39 years and 60–70 years, with 73.08% being in the age group between 40 and 49 years. Females made up of 76.92% and 23.08% were male. We also found that 67.31% of parents were married, 11.57% divorced, 9.61% lived in a relationship, 7.69% were single and 3.84% were widowed. To finish the sociodemographic characterization of the parents, we questioned them as to their level of education and found that 44.23% have a degree, 38.46% have a secondary education, 11.54% have a master’s degree, 3.84% have basic education and only 1.92% have a doctorate.

### 3.3. Results of Minors in Different Evaluation Measures

#### 3.3.1. Understanding of the Child Regarding the Information Provided by the Physician: QuIC Part A

Part A of the QuIC is composed of 17 statements and each has three response options: Disagree, with score 0; I am not sure, with a score of 1; and I agree, with a score of 2.

Through the results presented in Table 1, we can see that most of the children have a high ability to understand the objective information provided by the doctor when requesting informed assent. The values presented are very close to the maximum value of the scale, except for question A11: “There is a possibility that the stress echocardiography does not clarify the existing doubts about my possible health problem”, in which, despite presenting a positive mean which was 1.4 out of a total of 2, it has a mean lower than that of most questions. Thus, we can say that in this question, the children had some difficulty in understanding the diagnostic capacity of the stress echocardiography.

We also highlighted question A13 “The assent form that I signed lists the names of the researchers of this study, which I can contact if I have questions or concerns about my participation in this study”. This question has the lowest mean, although positive (1.13 out of a total of 2).

In contrast, question A1 “When I signed the assent form to perform the stress echocardiography, I knew I was agreeing to participate in a research study”, with a mean of 2 in 2 shows the total understanding by the children regarding their inclusion in the study.

We also highlight the answers to question A16 “The information given by the doctor who performed the stress echocardiography helped me to be calmer” with a mean of 1.6 in 2; furthermore, the question A17 “I felt more adult and responsible because the doctor asked me to perform the stress echocardiography” had a mean of 1.53 out of a total of 2. These results demonstrate the importance that children attribute to the fact that they were asked to give their assent (Table 1).

#### 3.3.2. QuIC Part B

Part B of the QuIC is composed of 14 statements and each has five response options: I did not understand this at all, scored with 1; I understood this very little, scored with 2; I understood this a little, scored with 3; I understood this well, scored 4; and I understood this very well, scored with 5.

From the descriptive analysis, shown in Table 2, we can see that all questions had values higher than 4, which shows an excellent ability of children to understand subjective information. We emphasize question B5, “Which of the procedures is experimental”, for which the mean is less than 4 although it still has a positive value (3.25).

Of note is the high capacity of understanding shown by the children in more subjective questions related to their participation in research, namely, the risks, discomfort and associated benefits or even the potential benefit to other children resulting from their participation in this study.

This ability to understand their involvement in the study is unequivocally expressed in question B1 “The performance of your stress echocardiography was used for a research”, which had a mean of 4.63 out of 5; in question B2 “Why the researchers are doing this study”, which had a mean of 4.5 out of 5; and in question B3 “How long you will be participating in this study”, which had a mean of 4.1, of a total of 5.

We also highlighted question B14 “In its entirety, how well did you perceive the study when you signed the assent form”, with a mean of 4.47 out of a total of 5, showing that a high percentage of children perceived the study and the various aspects related to it when signing the assent form.

The results of question B12 “Who can you contact if you have questions or concerns about your participation in this study” are less positive, which confirms what was previously shown in the analysis of QuIC part A, where a greater effort was necessary to identify the researchers (Table 2).

To better understand the breadth of the results of the two parts of the questionnaire, we present the mean sum of the QuIC responses in both parts. Thus, concerning QuIC Part A, we found that the mean sums are 29.76 out of a total of 34, which corresponds to 85% of the total sum of the values; regarding QuIC Part B, we found that the mean sum of the responses is 62.17 out of a total of 70, which corresponds to 88.81% of the total.

These values are very elucidative of the ability of children to understand both objective and subjective information.

To evaluate whether the older group (≥14 years old) had better comprehension ability than the younger group (<14 years old), we performed a Welch *t*-test. From the analysis of the results, we found that there are no statistically significant differences between younger children (10 years old and <14 years old) and older children (14 years old and <18 years old) regarding their ability to understand the information provided by the physician in the context of the assent. Regardless of age, all minors had a high ability to understand the information (Table 3).

A Pearson correlation was also performed to assess whether there is a relationship between both parts of the QuIC. The results show a positive linear correlation between the two parts of the questionnaire: r = 0.576 (*p* < 0.001), which means that the children who scored more in part A were simultaneously those who scored more in part B.

To assess the relationship between the items’ sum in part A and B of the children’s questionnaire with the total of their respective parents, we considered a correlational analysis via Pearson’s correlation coefficient. However, there was no relationship between the scores of their parents with the children’s total obtained in part A (r = 0.139; *p*-value = 0.327) and B (r = 0.142; *p*-value =0.315) of their questionnaires.

### 3.4. Results of Parents in Different Evaluation Measures

#### Parents’ Opinion Related to the Importance of the Assent Application to the Minor

From the analysis of Table 4, we can see that a high percentage of parents agree that the physician’s performance was very positive. Thus, 96.2% considered that the doctor introduced himself to the minor; 96.2% considered that the information regarding the exam was transmitted in a language appropriate to his age/maturity; 94.2% considered that the doctor explained to the child the procedures he would perform and tried to live up to his expectations; 80.8% considered that the doctor explained the possible discomfort to the child, highlighting in this question a greater number of parents/tutors who reported that this did not happen, which enabled us to realize the flaws that health professionals still commit in what relates to the explanations of clinical acts and when related to the child. In general, we gave more value to the informed consent requested from parents, to whom we provide more detailed and appropriate information. The same happens with question 5 “Explained to the child the possible side effects that may occur during the exam” (where 25.0% of parents/tutors reported that the doctor did not address the possible side effects) and question 6 “Explained to the minor the possible risks, as well as the necessary actions to minimize or correct them” (23.1% of parents considered that the doctor did not explain to the minor the possible risks and the necessary actions to correct them). These questions highlight what was expressed above, which is the importance given by health professionals to the informed consent provided by parents/tutors to whom all information is made available and not valuing the availability of that information to minors, often due to the fear that so much information can create anxiety and fear.

Question 7 “Explained to the child the benefits that the exam will bring to their health” shows that 88.5% of the parents considered that the doctor explained the benefits of the exam; and, finally, question 8 “From the above, I consider the application of the informed assent an asset for the acceptance of the exam, by the minor”, which demonstrates, without a doubt, the very positive opinion of parents/tutors (98%) on the importance of applying informed assent for the minor to accept the examination (Table 4).

## 4. Discussion

The results of this study show that minors can understand the information given to them by health professionals. However, where children are concerned, we do not know when exactly they can decide on medical issues in a meaningful manner [24].

Our findings show that some children have difficulty in understanding a few aspects related to diagnostic procedures and other aspects of the study procedures. In this sense, Roth-Cline and Nelson [9] argue that the amount of information that a child must comprehend to provide meaningful and developmentally appropriate child assent (or dissent) should be allowed to vary with the age and maturity of the child. Thus, the child’s inability to understand otherwise important informational elements of informed consent, such as any reasonably foreseeable risks, does not establish that a child is incapable of agreeing or disagreeing to research participation [9].

The existing literature demonstrates the difficulty in deciding clearly from what age a minor can make valid decisions. In this sense, Grootens-Wiegers et al. [25] point that in the clinical research context children of 11.2 years and above were generally competent. In the treatment context, initial indications point into the direction of comparable age limits for alleged competence, which is around the age of 12.

The comprehension capacity is directly related to the manner by which the health professional transmits the information to the child. In this case, Burke et al. [26] emphasized that if the information is adapted to the age of the children, they can understand potentially difficult and complex concepts regarding their participation in biomedical research from the age of 6 years old.

Based on this consideration of age as a decisive factor for assenting, we present the results of another study for which its objective was to evaluate the key factors for the child’s competence to consent to participate in clinical research studies, which consisted of a sample of 161 children aged between 6 and 18 years old. The results showed a strong correlation between competence to decide and age [27].

The skills necessary to understand relevant information are usually acquired at 11 or 12 years of age when the individual’s thinking begins to function within models of verbal ideation and becomes capable of performing abstraction and logical operations [1]. These findings agree with the results obtained in our study, which showed a high ability to understand the information made available to them with a positive relationship between both forms of information, i.e., the greater the ability to understand objective information, the greater the ability to understand subjective information.

A study conducted by Grooten-Wiegers et al. (2017) with minors aged between 12 and 17 years old on their perspective regarding their involvement in research found that all the participants showed a good ability to understand [25].

Another study conducted by Grady et al. (2014) aimed at understanding the importance given by adolescents to their involvement in the decision making process found that 98% of the 177 adolescents between 13 and 17 years of age in their study reported wanting to sign an informed assent form and that it was easy to read and understand what they received in writing [28].

A previous study by Hein et al. (2015) reported similar findings as to that of our own reported here [29].

In sum, it can be concluded that there are no statistically significant differences between the two groups of children (10 years old and <14 years old and 14 years old and <18 years old) regarding their ability to understand the information. The children aged between 10 and 17 years showed high understanding ability related to the information that was made available to them to obtain assent to the medical procedure. However, there is the possibility that some children answered the questions based on what adults expected from them. Nonetheless, they considered their involvement in decision making a determining factor for reducing anxiety and increasing the sense of maturity and responsibility. There are no statistically significant differences between the female children compared to the male children regarding the ability to understand the information made available to them in the context of assent. The parents considered the implementation of the assent as a fundamental step to the child’s acceptance of health care.

## 5. Conclusions

Based on findings from previous studies as well as our own, we can conclude that decision making competence is a process and not a fixed phenomenon. Therefore, the confirmed potential for competence combined with the influence of other factors affecting it enables us to recommend a double consent procedure (child and parent) for minors from the age of 12 until 18. A double consent procedure will do justice for both developmental aspects of children and the specific characteristics of the parent-child dyad [25].

Thus, the literature shows that the ability of minors to assent is not based only on age but also on the ability of the child to understand and weigh the options and consequences of their involvement in the process and other factors, such as the objective of the study, the methods to be used and the type of information to be collected, may interfere with the decision making ability of the child and may have a weight greater than that of their age [30].

As a recommendation resulting from these results, we emphasize the need and relevance of the widespread application of child assent models.

As limitations, we emphasize that this is a convenience sample of small size. To expand our findings, we would need a large sample size (which we did not have) followed by an exploratory factor analysis.

The measurement instrument was not originally developed for use with children, which prevents us from finding studies that can serve as a comparison for the present study.

Cronbach’s alpha is an index of the consistency of a scale-reliability, i.e., it refers to the degree to which a scale correlates with its own “true” score and uncorrelates with extraneous random factors. A “high” value for alpha does not imply that the measure is unidimensional. In our case the internal consistency of the items is acceptable because α is over 0.65. To provide evidence that the scale in question is unidimensional, an exploratory factor analysis must be considered.

There was no objective validation of the understanding demonstrated by the minors regarding the information made available to them, for example, through an interview where they had to describe what was explained to them.

Another limitation might be that the same version of the QuIC was administered to the whole range of children aged 10 to 17.

### Practice Implications

Improving the knowledge of health professionals about the ability of minors to understand the information made available to them can improve children’s participation in healthcare situations and contribute to accomplishing true child and family centered care.

Health professionals must inform parents of the benefits of involving minors in the decision making process. Therefore, they should encourage the adoption of a more open communication style by involving the whole family.

Parents should be encouraged to participate in the decision making process involving assent because of their enhanced ability to interpret the behavior and statements of their children and to better define whether they will expresses assent or dissent.

Parents play an important role in helping children recognize their abilities and responsibilities as key actors in the actual assent process.

Depending on the child’s reading ability and comprehension, the researcher may read the form to or with the child. The child must be given the opportunity to ask questions and the researcher should have the opportunity to explain anything that is not clear.

For older minors, a written assent form is desirable. The researchers should consider the children’s reading level and adapt the language accordingly. It may require more concise explanations or the use of pictures or videos to better explain the study procedures.

## Figures and Tables

**Table 1 healthcare-09-00871-t001:** Responses related to the assessment of objective information understanding (QuIC Part A).

QuIC Part A	N	Min	Max	Mean	Mode	S.D.	%
A1. When I signed the assent form to perform the stress echocardiography, I knew I was agreeing to participate in a research study.	52	2	2	2.00	2	0.000	0
							0
							100.0
A2. The main reason for clinical studies is to improve the quality of care provided in the future.	52	1	2	1.97	2	0.183	0
							1.9
							98.1
A3. I was informed of the duration of my participation in this clinical study.	52	0	2	1.77	2	0.568	3.8
							9.6
							86.5
A4. All procedures in this study are appropriate to answer questions about my possible health problem.	52	1	2	1.83	2	0.379	0
							17.3
							82.7
A5. One of the main objectives of the researchers of this study is to create an assent model that can be applied in the performance of stress echocardiography in children.	52	0	2	1.77	2	0.568	3.8
							5.8
							90.4
A6. One of the main objectives of the researchers of this study is to verify whether the assent model used is adequate to perform stress echocardiography.	52	1	2	1.87	2	0.346	0
							9.6
							90.4
A7. One of the main objectives of the researchers of this study is to verify if I understood the information given to me about the possible risks of performing this test.	52	0	2	1.73	2	0.583	3.8
							13.5
							82.7
A8. Stress echocardiography is considered the safest and least risky test in the study of my possible health problem.	52	1	2	1.77	2	0.430	0
							25.0
							75.0
A9. The need to perform the stress echocardiography was decided by my doctor due to my possible health problem.	52	0	2	1.87	2	0.434	1.9
							7.7
							90.4
A10. Compared with alternative complementary tests (Coronary Angio CT, myocardial perfusion scintigraphy or catheterization), the stress echocardiography presents fewer risks or discomfort to my health.	52	1	2	1.83	2	0.379	0
							15.4
							84.6
A11. There is a possibility that the stress echocardiography does not clarify the existing doubts about my possible health problem.	52	0	2	1.40	2	0.675	9.6
							44.2
							46.2
A12. By participating in this clinical study, I am helping researchers acquire information that may benefit other children in the future.	52	0	2	1.87	2	0.434	1.9
							3.8
							94.2
A13. The assent form that I signed lists the names of the researchers of this study, which I can contact if I have questions or concerns about my participation in this study.	52	0	2	1.13	2	0.900	34.6
							17.3
							48.1
A14. If I did not want to participate in this clinical study, I could have refused to sign the assent form.	52	0	2	1.83	2	0.531	3.8
							1.9
							94.2
A15. The doctor who performed the stress echocardiography explained to me, in clear and simple language, what the test consists of and its importance for my diagnosis and treatment.	52	0	2	1.77	2	0.626	5.8
							7.7
							86.5
A16. The information given by the doctor who performed the stress echocardiography helped me to be calmer.	52	0	2	1.60	2	0.724	9.6
							11.5
							78.8
A17. I felt more adult and responsible because the doctor asked me to perform the stress echocardiography.	52	0	2	1.53	2	0.819	15.4
							3.8
							80.8

**Table 2 healthcare-09-00871-t002:** Responses related to the evaluation of the understanding of subjective information (QuIC Part B).

QuIC Part B	N	Min	Max	Mean	Mode
B1. The performance of your stress echocardiography was used for a research.	52	1	5	4.63	5
B2. Why the researchers are doing this study.	52	1	5	4.50	5
B3. How long you will be participating in this study.	52	1	5	4.10	5
B4. The procedures to which you will be subjected.	52	1	5	4.27	5
B5. Which of the procedures is experimental?	52	1	5	3.21	5
B6. The possible risks and discomfort of participating in this study.	52	1	5	4.10	5
B7. The possible benefits for you of participating in this study.	52	1	5	4.50	5
B8. How your participation in this study will benefit other children in the future.	52	2	5	4.63	5
B9. The alternatives to your participation in this study (performing the test and not participating in the study).	52	1	5	4.47	5
B10. The guarantee of the confidentiality of your participation in the study.	52	2	5	4.73	5
B11. Not having to pay any amount for your participation in this study.	52	2	5	4.83	5
B12. Who can you contact if you have questions or concerns about your participation in this study?	52	1	5	4.00	5
B13. That your participation in this study is voluntary.	52	4	5	4.90	5
B14. In its entirety, how well did you perceive the study when you signed the assent form.	52	3	5	4.47	5

**Table 3 healthcare-09-00871-t003:** Results of the Welch test to compare the responses of QuIC Part A and Part B between the two sample groups.

QuIC	Statistic	DF1	DF2	Sig.
Sum QuIC Part A Welch	0.391	1	31.903	0.536
Sum QuIC Part B Welch	0.009	1	39.084	0.926

**Table 4 healthcare-09-00871-t004:** Responses related to parents/tutors’ opinion about the importance of assent.

Parents/Tutors’ Opinion about the Importance of Assent	YES		NO	
	N	%	N	%
The doctor introduced himself to the child/adolescent:	50	96.2	2	3.8
2. Explained to him, in a clear and appropriate language for the maturity of the child/adolescent, what the exam consists of and its importance for its diagnosis and treatment:	50	96.2	2	3.8
3. He explained to the child/adolescent the procedures that will be performed, always trying to meet the expectations of the child/adolescent:	49	94.2	3	5.8
4. He explained to the child/adolescent the possible discomforts that may occur during the exam:	42	80.8	10	19.2
5. Explained to the child/adolescent about the possible side effects that may occur during the exam:	39	75	13	25
6. Explained to the child/adolescent the possible risks, as well as the necessary actions to minimize or correct them:	40	76.9	12	23.1
7. He explained to the child/adolescent the benefits that the exam will bring to his health:	46	88.5	6	11.5
8. From the above, I consider the application of the informed consent an added value for the acceptance of the exam by the child/adolescent:	50	98	1	2

## Data Availability

The data presented in this study are available upon request from the corresponding author. The data are not publicly available due to privacy reasons.

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
