# Peer review of "Children’s Understanding of Informed Assents in Research Studies"

_healthcare, 2021, doi:10.3390/healthcare9070871_

Round 1

Reviewer 1 Report

This study is interesting because of its emphasis on identifying children's understanding of assent procedures in research studies. Its focus on differentiating between younger and older children in their understanding is important, particularly as it reminds the reader that not all children are the same, or that "one size fits all"  when it comes to understanding what is being asked of them.  Inherent in the study is a nod to the value of children having a say in decision making as well as posing the question of when are they really able to make a decision that they truly understand. In this case, at what age do children truly understand when they give their assent to participate in a research study, that researchers can be fully assured of in conducting research with children.

Here are some suggestions and recommendations for your consideration. They are primarily to provide greater clarity and strengthen the points made by the authors.

Consider changing the name of the manuscript. It does not fully capture the main essence of the study and topic being investigated.  Consider something like Children’s Understanding of Informed Assents in Research Studies.

Lines 154-159, 178-180: sections could be combined and rationale provided as to why this particular group of children and parents selected for this study other than they were a convenient sample; Was there another reason for selecting these children such as the health condition?

Since the children ranged from 10 to 17 years of age were included in the study, was there any formal (intellectual capacity? reading and understanding ability or capacity)  or informal way that children were assessed “to be conscious and present cognitive ability to perceive the information provided”  (lines 179-180)?

It would provide greater clarity  to the reader if a heading was used that specifically described Consent procedures. Section 2.2-lines 160-177 describe consent procedures, whereas lines 154-159 and lines 178-180 describe the sample,  sample selection, inclusion and exclusion criteria. 

Similarly, a separate section/heading devoted to describing the measure, its psychometric properties is recommended; along with that, it is suggested that  the Procedures section be renamed Data Collection Procedures where only data collection procedures are described, and a data analysis section/heading be added describing the data analysis procedures.

Lines 186-188, clarify and give example of what is “objective information” and what is “subjective information” .  It seems that objective information refers to knowledge of study procedures and their benefits and risks shared by the doctor prior to asking for assent; subjective information refers to  understanding of possible impacts on children if they participated in the study.

Line 189,  Consider changing the sentence to read:  This is a brief questionnaire originally  developed to measure subjects’ actual (objective) and 189 perceived (subjective) understanding of consent procedures by adult cancer patients  participating in clinical trials.

Line 193-195,  Consider changing the sentence to read: To evaluate the internal consistency of this measure,  we calculated Cronbach’s alpha separately for part A and part B of the QuIC.    For part A and B of QuIC, the values obtained were 0.68 and 0.77, respectively.

Line 207, Consider changing the heading to: 3.1. Sociodemographic characteristics of minors

Lines 213-217, Consider changing the sentence to read: Crosstabulations between children’s age (< 14 and > 14)  and gender (male/female) revealed that 11.5% (n=6) of the children were female and 26.9% (n=14) were male under 14 years of age; 17.3% (n=9) were female and  44.2% (n=23) male equal to or over 14 years of age.  

Line 218, consider changing the heading to: 3.2. Sociodemographic characteristics of parents

Lines 230-231,  This statement should be stated earlier under the measure section as a part of describing how the measure is scored, can be repeated here as a reminder to the reader on how to interpret the results.

Lines 299-300 (Table 3). Review how results for Welch test  statistic are to be displayed on a in table format and make changes if necessary.

Line 356-357, consider changing it to say, These findings agree with the results obtained in our study

Line 361- consider changing the sentence to say:  A study conducted by Grooten-Wiegers  et al. with minors aged between 12 and 17 years old on their perspective regarding their involvement in research, found that all the participants  showed a good ability to understand.

Line 369-370, consider changing the sentence to say: Another study conducted by Grady et al.,(2014) aimed at understanding the importance given by  adolescents to their involvement in the decision-making process found that 98% of the 177 adolescents between 13 and 17 years of age in their study reported wanting to sign an informed assent form, and that it was easy to read and understand, which they received in writing.

Line 374,  consider changing the sentence to say:  A previous study by Hein et al. (2015) reported similar findings as to that of our own reported here.

 Line 379,  consider changing the 1st part of the sentence to say: Based on findings from previous studies as well as our own, we can conclude that….

Line 381, consider changing the statement to say:  leads us to recommend …

Lines 391-402, consider moving this section (lines 391-402) to be under discussion/summary of results.

Lines 379-390, consider moving this section (lines 379-390) to the conclusion section.

Lines 406-409,  this section should  be expanded in regards to specifically what needs to be improved in regards to knowledge of assent procedures, what education can be provided to parents about assent procedures? What further education can be provided to children on assent procedures? What ways assent procedures can be presented to children and their parents?

The limitations of the study need to be expanded and elaborated on.  What are the limitations of the study findings given that the measurement instrument was adapted and not originally developed for use with children? Additionally, the internal consistency of the measure was average, what are the implications of this for the validity and reliability of the results reported?

Were any data analysis considered exploring the relationship between children's responses to parental responses? 

Author Response

ANSWER REVIEWER 1

Dear Reviewer 1

I greatly appreciate all your suggestions, with which I have agreed and tried to respond in the text of the article.

In the text, I did some changes that the other reviewers asked me. I hope you agree with them.

Comments and Suggestions for Authors

This study is interesting because of its emphasis on identifying children's understanding of assent procedures in research studies. Its focus on differentiating between younger and older children in their understanding is important, particularly as it reminds the reader that not all children are the same, or that "one size fits all"  when it comes to understanding what is being asked of them.  Inherent in the study is a nod to the value of children having a say in decision making as well as posing the question of when are they really able to make a decision that they truly understand. In this case, at what age do children truly understand when they give their assent to participate in a research study, that researchers can be fully assured of in conducting research with children.

Here are some suggestions and recommendations for your consideration. They are primarily to provide greater clarity and strengthen the points made by the authors.

 Consider changing the name of the manuscript. It does not fully capture the main essence of the study and topic being investigated.  Consider something like Children’s Understanding of Informed Assents in Research Studies.

I agree with your suggestion of the title which I have changed.

Lines 154-159, 178-180: sections could be combined and rationale provided as to why this particular group of children and parents selected for this study other than they were a convenient sample; Was there another reason for selecting these children such as the health condition?

The reason for choosing these participants is their health condition, as they are children who are healthy only with some symptoms during the practice of physical exercise, which guarantees us the reduction of other confounding factors.

Since the children ranged from 10 to 17 years of age were included in the study, was there any formal (intellectual capacity? reading and understanding ability or capacity)  or informal way that children were assessed “to be conscious and present cognitive ability to perceive the information provided”  (lines 179-180)?

No. the doctor who performed the echocardiogram by providing all the information related to the examination and their inclusion in research, performed an informal assessment of the child's ability to understand and decided if the minor can be enrolled in the study.

It would provide greater clarity  to the reader if a heading was used that specifically described Consent procedures. Section 2.2-lines 160-177 describe consent procedures, whereas lines 154-159 and lines 178-180 describe the sample,  sample selection, inclusion and exclusion criteria.

I did all the suggestions you did. I hope they are correct.

Similarly, a separate section/heading devoted to describing the measure, its psychometric properties is recommended; along with that, it is suggested that  the Procedures section be renamed Data Collection Procedures where only data collection procedures are described, and a data analysis section/heading be added describing the data analysis procedures.

I did all the suggestions you did. I hope they are correct.

Lines 186-188, clarify and give example of what is “objective information” and what is “subjective information” .  It seems that objective information refers to knowledge of study procedures and their benefits and risks shared by the doctor prior to asking for assent; subjective information refers to  understanding of possible impacts on children if they participated in the study.

In the paper I added:

Objective information refers to the knowledge of study procedures and their benefits and risks shared by the doctor prior to asking for assent; subjective information refers to the understanding of possible impacts that participating in the study may have on children.

This is a brief questionnaire originally developed to measure subjects’ actual (objective) and perceived (subjective) understanding of consent procedures by adult cancer patients participating in clinical trials.

Line 189,  Consider changing the sentence to read:  This is a brief questionnaire originally  developed to measure subjects’ actual (objective) and 189 perceived (subjective) understanding of consent procedures by adult cancer patients  participating in clinical trials.

I did this change as you asked.

Line 193-195,  Consider changing the sentence to read: To evaluate the internal consistency of this measure,  we calculated Cronbach’s alpha separately for part A and part B of the QuIC.    For part A and B of QuIC, the values obtained were 0.68 and 0.77, respectively.

I did this change as you asked.

Line 207, Consider changing the heading to: 3.1. Sociodemographic characteristics of minors

I did this change as you asked.

Lines 213-217, Consider changing the sentence to read: Crosstabulations between children’s age (< 14 and > 14)  and gender (male/female) revealed that 11.5% (n=6) of the children were female and 26.9% (n=14) were male under 14 years of age; 17.3% (n=9) were female and  44.2% (n=23) male equal to or over 14 years of age. 

I did this change as you asked.

Line 218, consider changing the heading to: 3.2. Sociodemographic characteristics of parents

I did this change as you asked.

Lines 230-231,  This statement should be stated earlier under the measure section as a part of describing how the measure is scored, can be repeated here as a reminder to the reader on how to interpret the results.

I did this change as you asked.

Lines 299-300 (Table 3). Review how results for Welch test  statistic are to be displayed on a in table format and make changes if necessary.

I added more results to the table.

Line 356-357, consider changing it to say, These findings agree with the results obtained in our study

I did this change as you asked.

Line 361- consider changing the sentence to say:  A study conducted by Grooten-Wiegers  et al. with minors aged between 12 and 17 years old on their perspective regarding their involvement in research, found that all the participants  showed a good ability to understand.

I did this change as you asked.

Line 369-370, consider changing the sentence to say: Another study conducted by Grady et al.,(2014) aimed at understanding the importance given by  adolescents to their involvement in the decision-making process found that 98% of the 177 adolescents between 13 and 17 years of age in their study reported wanting to sign an informed assent form, and that it was easy to read and understand, which they received in writing.

I did this change as you asked.

Line 374,  consider changing the sentence to say:  A previous study by Hein et al. (2015) reported similar findings as to that of our own reported here.

I did this change as you asked.

 Line 379,  consider changing the 1st part of the sentence to say: Based on findings from previous studies as well as our own, we can conclude that….

I did this change as you asked.

Line 381, consider changing the statement to say:  leads us to recommend …

I did this change as you asked.

Lines 391-402, consider moving this section (lines 391-402) to be under discussion/summary of results.

I did this change as you asked.

Lines 379-390, consider moving this section (lines 379-390) to the conclusion section.

I did this change as you asked.

Lines 406-409,  this section should  be expanded in regards to specifically what needs to be improved in regards to knowledge of assent procedures, what education can be provided to parents about assent procedures? What further education can be provided to children on assent procedures? What ways assent procedures can be presented to children and their parents?

I added this information in the text of the paper:

Health professionals must inform parents of the benefits of involving minors in the decision-making process. Therefore, they should encourage the adoption of a more open communication style, involving the whole family.

Parents should be encouraged to participate in the decision-making process involving assent because of their enhanced ability to interpret the behaviour and statements of their children, and better defining whether their will expresses assent or dissent.

Parents play an important role in helping children recognize their abilities and responsibilities as key actors in the actual assent process.

Depending on the child’s reading ability and comprehension, the researcher may read the form to or with the child. The child must be given the opportunity to ask questions and the researcher should have the opportunity to explain anything that is not clear.

For older minors, a written assent form is desirable. The researchers should consider the children’s reading level and adapt the language accordingly. It may require more concise explanations, or the use of pictures or videos to better explain the study procedures.

The limitations of the study need to be expanded and elaborated on.  What are the limitations of the study findings given that the measurement instrument was adapted and not originally developed for use with children? Additionally, the internal consistency of the measure was average, what are the implications of this for the validity and reliability of the results reported?

I added this information in the text of the paper:

As limitations, we emphasize that this is a convenience sample of a small size. To expand our findings, we would need a large sample size (which we didn't have) followed by exploratory factor analysis.

The measurement instrument was not originally developed for use with children, which prevents us from finding studies that can serve as a comparison for the present study.

Cronbach’s alpha is an index of the consistency of a scale - reliability – i.e. it refers to the degree to which a scale correlates with its own “true” score and uncorrelates with extraneous random factors. A “high” value for alpha does not imply that the measure is unidimensional. In our case, the internal consistency of the items is acceptable because α is over .65. To provide evidence that the scale in question is unidimensional an exploratory factor analysis must be considered.

There was no objective validation of the understanding demonstrated by the minors regarding the information made available to them, for example, through an interview where they had to describe what was explained to them.

Another limitation might be that the same version of the QuIC was administered to the whole range of children aged 10 to 17.

Were any data analysis considered exploring the relationship between children's responses to parental responses?

I added this information in the text of the paper:

To assess the relationship between the items’ sum in part A and B of the children’s questionnaire with the total of their respective parents we considered a correlational analysis via Pearson’s correlation coefficient. However, there is no relationship between the scores of their parents with the children’s total obtained in part A (r=0.139; p-value=0.327) and B (r=0.142; p-value =0.315) of their questionnaires.

THANK YOU VERY MUCH FOR YOUR HELP IN IMPROVING MY PAPER.

Best regards

Hortense Cotrim

Reviewer 2 Report

The paper was a very interesting read and most definitely is of great value to the field. The authors report the study very clearly and it is very easy to read. It would be nice to see a few more similar studied described in the literature section, if there are any others, just to see more of a comparison and to highlight what new the current study brings to the field. A must read for health professionals. 

Author Response

ANSWER FOR REVIEWER 2

The paper was a very interesting read and most definitely is of great value to the field. The authors report the study very clearly and it is very easy to read. It would be nice to see a few more similar studied described in the literature section, if there are any others, just to see more of a comparison and to highlight what new the current study brings to the field. A must read for health professionals.

Answer:

Dear Reviewer, I am sending my answer to your suggestions for which I thank you very much.

The literature shows that children’s competence has never been systematically examined in a standardized manner which justifies the lack of other studies that we can use to compare our findings. However, we tried to include some studies that can help to improve the paper.

In the literature, studies to assess children's ability to understand are scarce as well as the assessment scales. Hein et al (2014) adapted for the pediatric population the MacCAT-CR scale initially developed for the adult population. Their study obtained results similar to the ones present in our study.

I hope that the changes made will meet your expectations.

Best regards

Hortense Cotrim

Reviewer 3 Report

Manuscript: Minor and Family Centred Care in Decision-Making Process (ID: healthcare-1239641)

The Authors present a study on the ethical question whether minors should be asked for assent in medical care. The authors are to be commented for the focus on children´s and minor´s rights and the consideration of their opinion in any treatment. Thus, this manuscript is valuable and interesting.

However, in my opinion the manuscript still lacks clarity that could be provided by some revisions and I want to ask the authors to address some major and some minor points.

Major points

Abstract

I would like to suggest to abandon numbers (e.g. correlation coefficients) from the abstract and concentrate on the main findings and instead give a very brief introduction on the context (medical care versus research) and why it may be important to obtain minors assent.

Abbreviations (e.g. QuIC) are not introduced and should be spelled out in the Abstract

Introduction

Please provide a clearer structure in the introduction. The first paragraph stated that children have the right to be heard and informed – and I very much agree. However, I did not understand the context – are you discussing essential and indispensable medical treatment and diagnostics or are you arguing the case for assent in research studies? In my opinion, the frame of the question needs to be set. Please elaborate this issue.

Furthermore, a quick introduction to “informed assent” as opposed by “informed consent” would be helpful right in the beginning of the text (it is briefly defined in line 100 etc pp but should be mentioned earlier)

After introduction of the context and a definition of “assent”, in my opinion, the (negative) consequences of non-consideration of minor´s assent should be presented (taking medical care versus research into account).

Results

In lines 232-234 it is stated that children “have high ability to understand the objective information”… In my opinion, the results show, that children feel well informed – but I will raise this point again in the discussion.

The results section should contain mainly numbers and tables, whereas the interpretations should be moved to the discussion. Lines 235-253 comprise mainly interpretations of the response behavior and should be moved to the discussion. In my opinion these are very valuable results and should be valued as such and discussed further. E.g. the observation that children appreciate being carefully informed is worth emphasizing (although maybe not too surprising – I find it important to note).

The observation that children obviously did not quite catch the scope of the diagnostic procedure (A11) and who to turn to in case of questions (A13) – also corresponding questions in Part B – may underpin my suspicion that the children may have not understood everything but rather answered many questions in a socially desired way. It would add value to the manuscript, if you discussed this matter further.

The results for Part B are also mixed with interpretation and discussion. All Item descriptions and conclusions drawn from the answers should be moved to the discussion.

Lines 312-326 should be moved to the discussion and highlighted. I find this very interesting! Maybe you can find some information on other European countries and the common strategies of involvement of children in the information procedure and discuss this in relation to your findings?

Discussion

If I understand correctly, the QuIC was the only instrument to assess the minors understanding of the study (and medical) related information. Although Part B is supposed to assess “objective information” (line 187), it seems still a subjective measure as it does not actually test the knowledge about the ongoing procedure but rather a subjective rating of (self-perceived) understanding. Please comment briefly in the discussion section. Furthermore, did you consider the possibility that effects of social desirability bias the answering patterns? Did you check for any kind of “objective” information on the level of understanding? If not – as I assume from the presentation – I would consider this an issue to discuss.

The discussion section should be abbreviated and more precise. In my opinion, the whole paragraph lines 354-377 could be removed as from your results it seems sufficiently clear that children at the age of 10 are capable of understanding when the information is presented in an adequate manner.

The interpretation and discussion of the results (as mentioned above) should be moved to the discussion.

The supplemental Tables were missing.

Minor points

Lines 34 and 91 the word “defend” seems an inappropriate here – please check if “argue” may be better suited in this context.

Line 127 “do questions” could be replaced by “ask questions” – please check.

Paragraph line 105 to 112 – I do not quite understand this section. Does it imply that parents should consent whereas minors should be asked for assent?

In the procedure section, lines 195-197 the quality criteria of the QuIC are presented – please add a reference.

Line 180 “as” should be “was”

Table 3 is only rudimentary and needs more information. Groups should be named, means and SD should be provided and (ideally) Effect sizes could be presented in addition to the groupmeans.

Lines 307 – the word “considered” is confusing. Do you mean “agree” or “endorse”?

Author Response

Answer to Reviewer 3

 Dear Reviewer

Thank you so much for your suggestions for improving the paper. However, I was not always able to respond to them, since it was necessary to respond to the suggestions of the three reviewers, and at the same time to maintain the congruence of the text.

Here, I try to explain most of the changes I did.

The Authors present a study on the ethical question whether minors should be asked for assent in medical care. The authors are to be commented for the focus on children´s and minor´s rights and the consideration of their opinion in any treatment. Thus, this manuscript is valuable and interesting.

However, in my opinion the manuscript still lacks clarity that could be provided by some revisions and I want to ask the authors to address some major and some minor points.

Major points

Abstract

I would like to suggest to abandon numbers (e.g. correlation coefficients) from the abstract and concentrate on the main findings and instead give a very brief introduction on the context (medical care versus research) and why it may be important to obtain minors assent.

Abbreviations (e.g. QuIC) are not introduced and should be spelled out in the Abstract

In this item, I did all the suggestions you asked me for, as you can confirm in the abstract.

Introduction

Please provide a clearer structure in the introduction. The first paragraph stated that children have the right to be heard and informed – and I very much agree. However, I did not understand the context – are you discussing essential and indispensable medical treatment and diagnostics or are you arguing the case for assent in research studies? In my opinion, the frame of the question needs to be set. Please elaborate this issue.

In this item, I added this text to improve the context of the study.

This study appears as a way of deepening ethical knowledge regarding the recognition for the decision-making capacity of minors concerning their participation in research studies, based on the notion of human vulnerability and the imperative need to create mechanisms that guarantee the protection of children, to ensure respect for their dignity as human beings and care for their best interests.

It is also considered that this study can contribute to the enhancement of the health professionals’ approach to vulnerable populations, in particular children, in the design of strategies conducive to their empowerment and involvement in the decision-making process.

The literature notes the importance of asking children for their assent, especially when they are enrolled in research. However, in order to comply with the best practices, assent should be requested whenever it is necessary to make a decision involving the health and well-being of the minor. In this study, we agreed with this assumption and argued that in research the assent is of utmost importance because frequently children do not benefit directly from the results obtained.

Furthermore, a quick introduction to “informed assent” as opposed by “informed consent” would be helpful right in the beginning of the text (it is briefly defined in line 100 etc pp but should be mentioned earlier)

This explanation was included in the text as:

Technically, the assent is a document that explains to the child in language s/he can understand the essence of what is planned in the research, as well as the fact that s/he can say ‘no’, or can change his/her mind midway through the research [3]. Accordingly, "Assent" is a term used to express willingness to participate in research by persons who are, by definition, too young to give informed consent but who are old enough to understand the proposed research in general, its expected risks and possible benefits, and the activities expected of them as subjects. Assent by itself is not sufficient, however. If assent is given, informed consent must still be obtained from the subject's parents or guardian.

"Informed consent" is the voluntary agreement of an individual, or his or her authorized representative, who has the legal capacity to give consent, and who exercises free power of choice, without undue inducement or any other form of constraint or coercion to participate in research. The individual must have sufficient knowledge and understanding of the nature of the proposed research, the anticipated risks and potential benefits, and the requirements of the research to be able to make an informed decision [8].

Baines (2011) argues that assent is emphasized in research but is largely ignored in medical treatment for children. So, child’s medical treatment may proceed with consent from the parents but the child’s participation in research requires both consent from the parents and the assent of the child and this may lead to a practical problem if the parent and child give incompatible responses when asked to consent and assent [17].

On the other hand, a double consent procedure will do justice to both the developmental aspects of children and the specific characteristics of the parent-child dyad. The parental role offers extra protection by creating the context for the child’s competent decision-making and by facilitating the child’s long-term autonomy. In general, the perspective and attitudes of the adults (both parents and clinician) towards the child may be an important predisposing factor in order to stimulate the highest competence in the child [18].

 After introduction of the context and a definition of “assent”, in my opinion, the (negative) consequences of non-consideration of minor´s assent should be presented (taking medical care versus research into account).

The literature is not clear in relation to this aspect: most studies mix the results from research and from clinical practice. In the text, we tried to justify this aspect using the results of this study.

Some studies argue that the lack of involvement can have negative consequences, such as increased fears and anxieties, reduced self-esteem, depersonalization, and lack of preparation for the procedures [15]. In the opinion of Coyne et al [15], children who are not involved may assume that their opinions are not important or relevant and may not seek to share them in the future.

In this sense, Rippen [16] argues the importance of being aware that children have the right to the best possible medical care, the best information and education in comprehensive language suited to their age and a patient-oriented approach. If these aspects are considered, children generally feel better when they become involved in health care decisions, and emotional problems and disorders in the development of the child can be prevented.

 Results

In lines 232-234 it is stated that children “have high ability to understand the objective information”… In my opinion, the results show, that children feel well informed – but I will raise this point again in the discussion. I said this affirmation because the measure used permit the evaluation of objective information.

We based it on the purpose of the measure, that was developed to measure the objective information (Part A) and Subjective information (Part B). However, we added a limitation in our study related to the aspect you point out:

There was no objective validation of the understanding demonstrated by the minors regarding the information made available to them, for example, through an interview where they had to describe what was explained to them.

The results section should contain mainly numbers and tables, whereas the interpretations should be moved to the discussion. Lines 235-253 comprise mainly interpretations of the response behavior and should be moved to the discussion. In my opinion these are very valuable results and should be valued as such and discussed further. E.g. the observation that children appreciate being carefully informed is worth emphasizing (although maybe not too surprising – I find it important to note).

In relation to the interpretations that were included in the results section, we agree with you, but we think that these explanations allow the reader to better understand our findings.

This is shown through this statement:

In this sense, Rippen [16] argues the importance of being aware that children have the right to the best possible medical care, the best information and education in comprehensive language suited to their age and a patient-oriented approach. If these aspects are considered, children generally feel better when they become involved in health care decisions, and emotional problems and disorders in the development of the child can be prevented.

The observation that children obviously did not quite catch the scope of the diagnostic procedure (A11) and who to turn to in case of questions (A13) – also corresponding questions in Part B – may underpin my suspicion that the children may have not understood everything but rather answered many questions in a socially desired way. It would add value to the manuscript if you discussed this matter further.

Our findings show that some children have difficulty in understanding a few aspects related to diagnostic procedures and other aspects of the study procedures. In this sense, Roth-Cline and Nelson (2013) argue that the amount of information that a child must comprehend to provide meaningful and developmentally appropriate child assent (or dissent) should be allowed to vary with the age and maturity of the child. Thus, a child’s inability to understand otherwise important informational elements of informed consent, such as any reasonably foreseeable risks, does not establish that a child is incapable of agreeing or disagreeing to research participation [25].

We also included this paragraph:

The children aged between 10 and 17 years showed a high understanding ability, related to the information that was made available to them to obtain assent to the medical procedure. However, there is the possibility that some children answered the questions based on what adults expected from them.

The results for Part B are also mixed with interpretation and discussion. All Item descriptions and conclusions drawn from the answers should be moved to the discussion.

Lines 312-326 should be moved to the discussion and highlighted. I find this very interesting! Maybe you can find some information on other European countries and the common strategies of involvement of children in the information procedure and discuss this in relation to your findings?

We didn’t find any studies that discussed the strategies used, but we found some ethical documents that highlight the use of records with active voice, videos or written letters that are read to children.

In our paper we wrote this paragraph:

For older minors, a written assent form is desirable. The researchers should consider the children’s reading level and adapt the language accordingly. It may require more concise explanations, or the use of pictures or videos to better explain the study procedures.

Discussion

If I understand correctly, the QuIC was the only instrument to assess the minors understanding of the study (and medical) related information. Although Part B is supposed to assess “objective information” (line 187), it seems still a subjective measure as it does not actually test the knowledge about the ongoing procedure but rather a subjective rating of (self-perceived) understanding. Please comment briefly in the discussion section. Furthermore, did you consider the possibility that effects of social desirability bias the answering patterns? Did you check for any kind of “objective” information on the level of understanding? If not – as I assume from the presentation – I would consider this an issue to discuss.

In the literature, we found another tool that was adapted by Hein et al (2012) for the pediatric population. This is the MacCAT-CR scale which is a semi-structured interview format that helps clinical investigators to assess research adult candidates’ competence to give informed consent to participation in trials.

In the opinion of Hein et al (2012), research on the MacCAT scales in children has been limited to two small studies. Both studies confirmed the feasibility of using the MacCAT scales for children, but neither tested their validity and reliability. More rigorous research is needed on the applicability of the MacCAT instruments for children.

The modified MacCAT-CR for Children and Adolescents was used in research, and we think that the QuIC can be used simultaneously in clinical practice and research.

As you can confirm, the Part B of the QuIC assessed subjective information. In the text we include the following explanation:

Objective information refers to the knowledge of study procedures and their benefits and risks shared by the doctor prior to asking for assent; subjective information refers to the understanding of possible impacts that participating in the study may have on children.

The discussion section should be abbreviated and more precise. In my opinion, the whole paragraph lines 354-377 could be removed as from your results it seems sufficiently clear that children at the age of 10 are capable of understanding when the information is presented in an adequate manner.

In relation to the interpretations that were included in the results section, we agree with you, but we think that these explanations allow the reader to better understand our findings.

The interpretation and discussion of the results (as mentioned above) should be moved to the discussion.

The supplemental Tables were missing.

Sorry, the tables were included in the text. The supplemental documents are now included.

Minor points

All the minor points were included.

Lines 34 and 91 the word “defend” seems an inappropriate here – please check if “argue” may be better suited in this context.

Line 127 “do questions” could be replaced by “ask questions” – please check.

Paragraph line 105 to 112 – I do not quite understand this section. Does it imply that parents should consent whereas minors should be asked for assent?

In the procedure section, lines 195-197 the quality criteria of the QuIC are presented – please add a reference.

Line 180 “as” should be “was”

Table 3 is only rudimentary and needs more information. Groups should be named, means and SD should be provided and (ideally) Effect sizes could be presented in addition to the groupmeans.

Lines 307 – the word “considered” is confusing. Do you mean “agree” or “endorse”?